

# Low genetic diversity in captive populations of the critically endangered Blue-crowned Laughingthrush (*Garrulax courtoisi*) revealed by a panel of novel microsatellites

Guoling Chen[1], Chenqing Zheng[2], Nelson Wan[3], Daoqiang Liu[4], Vivian Wing Kan Fu[5], Xu Yang[2], Yat-tung Yu[5] and Yang Liu[1]

[1] State Key Laboratory of Biocontrol, Department of Ecology and School of Life Sciences, Sun Yet-sen University, Guangzhou, Guangdong, China
[2] Shenzhen Realomics Biological Technology Ltd, Shenzhen, Guangdong, China
[3] Ocean Park Corporation Hong Kong, Aberdeen, Hong Kong S.A.R., China
[4] Nanchang Zoo, Nanchang, Jiangxi, China
[5] The Hong Kong Bird Watching Society, Kowloon, Hong Kong S.A.R., China

## ABSTRACT

**Background:** Captive populations permit research and conservation of endangered species in which these efforts are hardly implemented in wild populations. Thus, analysing genetic diversity and structure of captive populations offers unique opportunities. One example is the critically endangered Blue-crowned Laughingthrush, *Garrulax courtoisi*, which has only two known wild populations in Wuyuan, Jiangxi and Simao, Yunnan, China. We carried out the first conservation genetic study, in order to provide useful implications that allow for successful ex situ conservation and management of the Blue-crowned Laughingthrush.

**Methods:** Using the novel microsatellite markers developed by whole-genome sequencing, we genotyped two captive populations, from the Ocean Park Hong Kong, which are of unknown origin, and the Nanchang Zoo, which were introduced from the Wuyuan wild population since the year 2010–2011, respectively. The genetic diversity of captive Blue-crowned Laughingthrush populations was estimated based on genetic polymorphisms revealed by a new microsatellite data set and mitochondrial sequences. Then, we characterised the population structure using STRUCTURE, principal coordinates analysis, population assignment test using the microsatellite data, and haplotype analysis of mitochondrial data. Additionally, we quantified genetic relatedness based on the microsatellite data with ML-Relate.

**Results:** Our results showed equally low levels of genetic diversity of the two captive Blue-crowned Laughingthrush populations. The population structure analysis, population assignment test using the microsatellite data, and haplotype analysis of the mitochondrial data showed weak population structuring between these two populations. The average pairwise relatedness coefficient was not significant, and their genetic relatedness was quantified.

**Discussion:** This study offers a genetic tool and consequently reveals a low level of genetic diversity within populations of a critically endangered bird species. Furthermore, our results indicate that we cannot exclude the probability that

Corresponding author
Yang Liu, liuy353@mail.sysu.edu.cn

the origin of the Hong Kong captive population was the wild population from Wuyuan. These results provide valuable knowledge that can help improve conservation management and planning for both captive and wild Blue-crowned Laughingthrush populations.

## INTRODUCTION

Many endangered species require captive breeding to save them from extinction, as they are incapable of surviving in inhospitable natural environments because of direct or indirect human impacts in the form of habitat loss, overexploitation, pollution, or introduced predators, competitors, or diseases (*Frankham, 2008*; *Frankham, Ballou & Briscoe, 2010*). For conservation management of captive populations, it is necessary to understand some baseline genetic information, such as the genetic diversity and population structure, to maintain the genetic 'health' of species for long-term viability (*Frankham, Ballou & Briscoe, 2010*). Furthermore, captive populations provide ideal systems for ecological, evolutionary, and genetic research of endangered species, because it is easier and safer to obtain DNA samples and observe the ecological habit of endangered species (*Ballou & Foose, 1996*; *Frankham, 2010*; *Frankham, Ballou & Briscoe, 2010*). Therefore, it is both extremely important and efficient to use captive populations as a resource to elucidate the genetic status of endangered species, and to plan appropriate genetic management (*Frankham, 2008*; *Frankham, Ballou & Briscoe, 2010*).

Because of their relatively high level of polymorphism, repeatability in genotyping and high PCR amplification success, microsatellites are advantageous genetic tool to address conservation and behavioural genetic patterns in threatened species towards their conservation and management issues (*Faria et al., 2016*; *Wang et al., 2017*). A key part of microsatellite-based conservation genetics is to develop either species-specific or cross-species primer sets, with the development of next-generation sequencing, the development of reliable microsatellites are no longer an time-consuming task, even for birds that have few known microsatellite loci in their genomes (*Castoe et al., 2012*; *Wang et al., 2017*; *Yang et al., 2017*).

The Blue-crowned Laughingthrush, *Garrulax courtoisi*, is listed as 'Critically Endangered' by the IUCN Red Data Book (*BirdLife International, 2017*) and has an extremely restricted distribution in south-eastern China. The entire known wild population of the nominate subspecies, which consists of approximately 300 individuals, is restricted to six fragmented sites in Wuyuan County, Jiangxi Province, China (*He et al., 2017*). Historical records indicate a disjunct population in southern Yunnan Province, sometimes treated as a distinct subspecies *G. c. simaoensis* (*Cheng & Tang, 1982*; *He et al., 2017*). However, the current status of this population is not known, and it has not been encountered in the wild since 1956 (*Wilkinson & He, 2010a*). In addition,

around 200 captive Blue-crowned Laughingthrush individuals are kept in several zoos in China, Europe, and America without known subspecies origin (*Wilkinson & Gardner, 2011*; *Wilkinson et al., 2004*).

Most recent studies on this rare species focused on habitat use and ecology in the wild Wuyuan population (*He et al., 2017*; *Wilkinson & He, 2010a*; *Zhang et al., 2017*), breeding ecology of captive populations (*Liu et al., 2017*, *2016*; *Wilkinson et al., 2004*), and its taxonomic status and conservation management (*Cheng & Tang, 1982*; *Collar, Robson & Sharpe, 2019*; *Wilkinson & He, 2010b*, *2010c*). However, there is no information on the genetic diversity, population structure, or other important issues pertaining to conservation genetics in either the wild or captive Blue-crowned Laughingthrush populations. Elucidating genetic diversity is thus important for producing a better understanding of microevolutionary processes and developing appropriate conservation and management strategies of Blue-crowned Laughingthrush.

Here, we present the first conservation genetic analysis of Blue-crowned Laughingthrush using mitochondrial DNA and a novel set of microsatellite markers developed by next-generation sequencing using the Illumina high-throughput sequencing platform. We characterised the genetic diversity, genetic structure, and relatedness in the only two captive populations in China, namely, the Ocean Park Hong Kong (OPHK) and the Nanchang Zoo (NCZ). Specifically, we attempted to learn two aspects. First, whether the two captive populations have a low level of genetic diversity and a sign of inbreeding, as expected for an endangered species. Second, whether the introduced origin of the population in the OPHK was from Wuyuan wild population, using the population of the NCZ as a reference.

## MATERIALS AND METHODS

### Sample collection and DNA extraction

The OPHK population was introduced in 1989, and the existing population size is 16 individuals. Both the source and individual relationships of Blue-crowned Laughingthrush are unknown because of foot ring loss and lack of records. The NCZ population in Jiangxi was introduced in 2010 and 2011 from the wild Blue-crowned Laughingthrush population in Wuyuan County, Jiangxi Province, and is the only captive population that has a confirmed source in the world. In this population, six individuals are from the wild population and the other seven individuals are their descendants.

Since this study involved sampling of endangered species, all the animal operations were approved by the Institutional Ethical Committee of Animal Experimentation of Sun Yat-sen University and strictly complied with the ethical conditions by the Chinese Animal Welfare Act (20090606). And all sampling procedures were performed with assistance of veterinarians or zoo keepers.

We collected Blue-crowned Laughingthrush samples from 23 individuals of two captive populations: 14 individuals from the long-established OPHK population and nine individuals from the recently established NCZ population (Table 1). Fresh blood samples from the 14 OPHK individuals were obtained in a non-invasive manner during regular veterinary examinations, and muscle samples from dead individuals and three egg remains were obtained from the NCZ population. All samples were stored in 95% ethanol

**Table 1 Sample voucher numbers and sequence GenBank accession numbers of two captive Blue-crowned Laughingthrush (*Garrulax courtoisi*) populations used in this article.**

| Taxon | Locality | Sample ID/Voucher | Gender | GenBank accession number of cytb sequence |
|---|---|---|---|---|
| *Garrulax courtoisi* | | SYSb6027 | M | MH423582 |
| | | SYSb6028 | F | MH423583 |
| | | SYSb6029 | M | MH423584 |
| | | SYSb6030 | F | MH423585 |
| | | SYSb6031 | F | MH423586 |
| | | SYSb6032 | F | MH423587 |
| | | SYSb6033 | M | MH423588 |
| | OPHK, Hong Kong, China | SYSb6034 | M | MH423589 |
| | | SYSb6035 | M | MH423590 |
| | | SYSb6036 | M | MH423591 |
| | | SYSb6037 | M | MH423592 |
| | | SYSb6038 | F | MH423593 |
| | | SYSb6039 | M | MH423594 |
| | | SYSb6040 | M | MH423595 |
| | | SYSb6041 | M | MH423596 |
| | | SYSb6042 | F | MH423597 |
| | | SYSb6043 | F | MH423598 |
| | | SYSb6044 | M | MH423599 |
| | Nanchang Zoo, Jiangxi, China | SYSb6045 | F | MH423600 |
| | | SYSb6046 | M | MH423601 |
| | | SYSb6047 | Unknown | MH423602 |
| | | SYSb6048 | Unknown | MH423603 |
| | | SYSb6049 | Unknown | MH423604 |

**Note:**
OPHK, Ocean Park, Hong Kong; SYSb, Sun Yat-sen University.

at −80 °C. We extracted total genomic DNA using the QIAamp DNA Mini Kit (Qiagen, GmbH, Hilden, Germany) following the manufacturer's protocol, and quantified DNA quality with a NanoDrop ND-1000 (Thermo Fisher Scientific, Waltham, MA, USA).

## Genome sequencing, microsatellite loci identification, and primer design

We sequenced the whole genome from one sample of Blue-crowned Laughingthrush (No. SYSb6040) from OPHK, and then used it as the draft genome. P5 and P7 adapters were ligated to the fragments after the genomic DNA was digested. The P5 adapter contains a forwards amplification primer site, an Illumina sequencing primer site, and a barcode. The selected fragments were end-repaired and 3′ adenylated, and these fragments were PCR amplified with P5- and P7-specific primers. Our library was validated using the Agilent Technologies 2100 Bio-analyzer and ABI StepOnePlus Real-Time PCR

System. After adapter ligation and DNA cluster preparation, the samples were sequenced using a Hiseq X-10 sequencer (BGI, Shenzhen, China).

Raw data from this single individual were processed by removing adapter sequences and subsequently removing the reads. Sequences with a low-quality rate (quality value $\leq 5$ (E)) greater than or equal to 50% and with more than 10% unknown ('N') bases were removed. The final read length was trimmed to 82 nucleotides (minimum length). Then, the high-quality sequences were selected to assemble the reference scaffolds. The genome was assembled using a short-read assembly method in SOAPdenovo2 (*Li et al., 2010*). A de Bruijn graph was built by splitting the reads into K-mers from the short-insert libraries (270 bp) without using pairing information. After a series of graph simplifications, the reads were assembled into contigs. All available paired-end reads were realigned onto the contig sequences to infer linkage between contigs. The linkage was removed if it was supported by unreliable paired-end reads. To simplify the contig linkage graph, we used subgraph linearisation, which extracted information on unambiguously linear paths. Iterative scaffolding was carried out to estimate insert size. Finally, to fill the intra-scaffold gaps, a local assembly was performed to locate the reads in the gap region, thus ensuring the other end of a scaffold was uniquely mapped to the linked contig.

We identified the microsatellites by screening the sequence data for di-, tri-, tetra-, and penta-nucleotide motifs with a minimum of six, five, five, and five repeats, respectively, by using the polymorphism information from the Blue-crowned Laughingthrush draft genome. Then we designed the primers in MSATCOMMANDER v.1.0.8 (*Faircloth, 2008*) and Primer 3 (*Rozen & Skaletsky, 2000*) to minimize potential structural or functional defects. After these procedures, we randomly selected a panel of 20 novel di-nucleotide markers and 10 tri-nucleotide markers.

## Microsatellite genotyping

The 30 selected loci were arranged into eight PCR multiplex sets (two to four loci per set); each forwards primer was labelled with fluorescent dye on the 5′ end of the forwards primers, and the sequence GTTTCTT was placed on the 5′ end of the reverse primer (*Brownstein, Carpten & Smith, 1996*). PCR amplifications were performed in a reaction volume of 10 μL, containing, five μL 2× PCR mix (QIAGEN Multiplex Kit), two μL 5× Q-Solution, one μL of a primer mix, and one μL of template DNA. The cycling conditions were as follows: initial denaturation at 95 °C for 15 min, followed by 35 cycles of denaturation at 94 °C for 30 s, annealing at 58 °C for 90 s and at 72 °C for 90 s, and a final extension at 72 °C for 10 min. Products were isolated and detected on an ABI Prism 3730XL Genetic Analyzer (Applied Biosystems, Carlsbad, CA, USA), and the fragment lengths were determined against an internal size standard (GeneScan™ 500 LIZ Size Standard; Applied Biosystems, Carlsbad, CA, USA) with GeneMapper v.3.7 (Applied Biosystems, Carlsbad, CA, USA). All samples were genotyped at the 30 microsatellite loci that were developed from the Blue-crowned Laughingthrush draft genome. We ultimately selected 19 microsatellite loci for our study and discarded the remaining ones because of low polymorphism.

## Mitochondrial DNA sequencing

To infer maternal relatedness of sampled individuals, partial mitochondrial cytochrome b (cytb) sequences were amplified and sequenced using the primers L14995 and H16065 (*Groth, 1998*). PCR amplifications were performed in a 20-μL reaction volume that contained one to two μL template DNA (50–100 ng), 10 μL 2× buffer, two μL dNTPs (two mM), 0.5 μL MgCl$_2$ (2.5 mM), 0.5 μL of each primer (10 mM), and 0.5 μL (one unit/μL) KOD DNA polymerase (Toyobo, Osaka, Japan). The PCR cycling conditions were as follows: an initial denaturation step of 4 min at 94 °C followed by 35 cycles of 40 s denaturation at 94 °C, 40 s annealing at 56 °C, and 90 s extension at 72 °C, followed by a final 10 min extension at 72 °C. The purified products were sequenced with both forwards and reverse primers using a BigDye Terminator v.3.1 Cycle Sequencing Kit (Applied Biosystems, Carlsbad, CA, USA) according to the manufacturer's guidelines. The products were sequenced on an ABI Prism 3730 Automated DNA sequencer (Shanghai Majorbio Bio-pharm Technology Co., Ltd., Shanghai, China).

## Genetic diversity estimates

For each microsatellite locus, we calculated the frequency of null alleles using Cervus v.3.0.7 (*Kalinowski, Taper & Marshall, 2007*); then, we used Arlequin v.3.5 (*Excoffier & Lischer, 2010*) to further test for Hardy–Weinberg equilibrium using 1,000 permutations and pairwise linkage disequilibrium by performing 100,000 Markov chain steps. Based on these three tests, seven out of 19 loci were removed from the dataset. To obtain genetic diversity estimates, we calculated the number of different alleles ($N_A$), average allelic richness ($A_R$), observed heterozygosity ($H_O$), and expected heterozygosity ($H_E$) using the remaining 12 loci in GenAlEx v.6.5.1 (*Peakall & Smouse, 2012*). We also calculated the inbreeding index ($F_{IS}$) for each population and assessed the significance of this index based on 10,000 permutations in Arlequin v.3.5.

In addition, we carried out rarefication analysis using POWSIM v.4.0 (*Ryman & Palm, 2006*) to assess the statistical power of our microsatellite markers to detect levels of population differentiation and relatedness (*Liu, Keller & Heckel, 2011*). Using an estimated effective size ($N_e$) of 1,000 for the base population, we performed 1,000 runs and generated eight predefined levels of population differentiation ($F_{ST}$ = 0.001, 0.0025, 0.005, 0.01, 0.02, 0.025, 0.05, 0.075), with sample sizes, numbers of markers, and allele frequencies corresponding to the empirical data. The proportion of significant outcomes ($P < 0.05$) then corresponded to an estimate of power. The $H_O$ at each locus was tested for equal allele frequencies by both Pearson's traditional contingency chi-square and Fisher's exact tests. The information from all loci was then combined by summing the data from chi-square and Fisher's methods (*Ryman & Jorde, 2001*; *Ryman & Palm, 2006*).

## Genetic population structure

For the microsatellite dataset, we applied four methods to estimate genetic population structure. First, we calculated pairwise $F_{ST}$ between these two populations, and derived significance levels using 10,000 permutations in Arlequin v.3.5. Sequential

Bonferroni correction (*Rice, 1989*) was used to adjust the significance levels for multiple testing. Second, we further tested the genetic structure using the Bayesian clustering method in STRUCTURE v2.3 (*Falush, Stephens & Pritchard, 2003*; *Pritchard, Stephens & Donnelly, 2000*). Using the Bayesian admixture model with the correlated allele frequencies option, we performed 1,000,000 Markov chain Monte Carlo iterations, with the first 200,000 discarded as burn-in. We conducted 10 independent runs for each $K$-value, the possible number of genetic cluster ($K$ = 1–4) for the entire dataset. We then used Structure Harvester v.0.6.8 (*Earl & Vonholdt, 2012*) to identify the most likely number of genetic clusters based on the ad hoc statistics described in *Evanno, Regnaut & Goudet (2005)*, in which both L($K$), the posterior probability, Ln P(D) increased per $K$, and $\Delta K$ means the second order rate of change of the Ln P(D) with respect to the number of clusters were estimated and compared. The final results for individual memberships were visualised by bar plot in DISTRUCT v.1.1 (*Rosenberg, 2004*). Third, principal coordinates analysis (PCoA) with pairwise Euclidian distances was carried out in GenAlEx v.6.5.1 (*Peakall & Smouse, 2012*) to visualise genetic relationships among individuals. Last, the biplots of pairwise population assignment likelihood values was computed using gamete-based Monte Carlo resampling method with a threshold of 0.01 in GenAlEx v.6.5.1 (*Paetkau et al., 2004*; *Peakall & Smouse, 2012*). This method uses genotype likelihoods to assign the possible population origins of individuals and allows estimation of dispersal events (*Paetkau et al., 2004*).

For DNA sequence data, we aligned the mitochondrial sequences using the Clustal W algorithm (*Thompson, Higgins & Gibson, 1994*) in MEGA v.6.06 (*Tamura et al., 2013*) with default parameters. The alignment was checked and manually adjusted when needed. To estimate the level of genetic polymorphism, basic genetic polymorphism statistics, such as haplotype number (h), haplotype diversity (Hd), number of segregating sites (S), and nucleotide diversity ($\pi$), of each population were calculated in DnaSP v.5.10.1 (*Librado & Rozas, 2009*). Then, this gene was analysed by haplotype network analysis using the reduced median-joining method (*Bandelt, Forster & Rohl, 1999*) in PopART v.4.8.4 (*Leigh & Bryant, 2015*).

## Relatedness analysis

For all relatedness estimates, the individuals from the OPHK and NCZ populations were separately analysed. For each pair of individuals from the OPHK or NCZ population, we calculated the *Queller & Goodnight (1989)* estimator of relatedness ($R_{QG}$) using GenAlEx v.6.5.1 (*Peakall & Smouse, 2012*). The genetic relatedness coefficient is defined as the proportion of ancestral alleles that are shared between descendants (*Lynch & Walsh, 1998*).

Then, the maximum likelihood estimates of relatedness (R) was calculated in ML-Relate (*Kalinowski, Wagner & Taper, 2006*), and the likelihood of four relatedness categories (unrelated: $R = 0$; close kin (e.g. half-siblings, aunt–niece): $R = 0.25$; full-siblings: $R = 0.5$; parent–offspring: $R = 0.5$) was used to determine the proportion of a specific relatedness category. To assess the likelihood of a given relatedness category relative to the other

**Table 2 Genetic variability in two captive Blue-crowned Laughingthrush (*Garrulax courtoisi*) populations of the analysed mitochondrial cytochrome b and 12 microsatellite loci.**

| Population | Mitochondrial DNA | | | | | Microsatellites | | | | | | |
|---|---|---|---|---|---|---|---|---|---|---|---|---|
| | $n$ | $K$ | $N_H$ | $H \pm$ SD | $\pi \pm$ SD (%) | $n$ | $N_A \pm$ SD | $A_R \pm$ SD | $H_O \pm$ SD | $H_E \pm$ SD | $F_{IS}$ | $R_{QG}$ |
| OPHK | 14 | 1.06 | 4 | $0.38 \pm 0.11$ | $0.10 \pm 0.12$ | 14 | $2.50 \pm 0.80$ | $4.42 \pm 3.92$ | $0.36 \pm 0.27$ | $0.34 \pm 0.22$ | $-0.214$ | $-0.08 \pm 0.52$ |
| NCZ | 9 | 1.10 | 2 | $0.37 \pm 0.11$ | $0.11 \pm 0.17$ | 9 | $2.75 \pm 0.97$ | $5.42 \pm 4.10$ | $0.45 \pm 0.24$ | $0.44 \pm 0.22$ | $-0.283$ | $-0.13 \pm 0.29$ |
| Total | 23 | 1.11 | 5 | $0.38 \pm 0.09$ | $0.11 \pm 0.13$ | 23 | $3.25 \pm 0.97$ | $6.25 \pm 4.11$ | $0.41 \pm 0.25$ | $0.39 \pm 0.21$ | $-0.231$ | $-0.05 \pm 0.33$ |

**Note:**

The number of individuals for which mtDNA ($N_{mt}$) and microsatellites ($N_{mi}$) were analysed are shown. For mtDNA, the average number of nucleotide differences ($K$), number of haplotypes ($N_H$), haplotype diversity ($H \pm$ SD), and nucleotide diversity ($\pi \pm$ SD, in percent) were calculated. For microsatellites, the average number of different alleles ($N_A \pm$ SD), average allelic richness ($A_R \pm$ SD), mean observed heterozygosity ($H_O \pm$ SD), and mean expected heterozygosity ($H_E \pm$ SD) were quantified. The multilocus inbreeding coefficients ($F_{IS}$, none of the coefficients were significant) and average pairwise relatedness based on the Queller and Goodnight estimator ($R_{QG} \pm$ SD) are provided for each population, and values in bold indicate significant deviations from Hardy–Weinberg equilibrium after Bonferroni correction.

three categories, a likelihood ratio test using a 95% confidence level and 1,000 simulations was carried out in ML-Relate (*Kalinowski, Wagner & Taper, 2006*).

## RESULTS

### Genome sequencing, microsatellite loci identification, and primer design

The whole genome of the endangered Blue-crowned Laughingthrush was first assembled using the high-coverage (approximately 40×) sequence reads. After processing of the approximately 41.18 Gb of raw data and removal of ambiguous barcodes, about 39.1 Gb of clean data were retained. The assembly generated 1,089,819 scaffolds larger than 100 bp, with an N50 contig size of 1,297 bp and an N50 scaffold size of 4,548 bp. The microsatellite detection generated 70,310 markers with 31,216 di-nucleotide repeats, 21,195 tri-nucleotide repeats, 10,892 tetra-nucleotide repeats, and 7,007 penta-nucleotide repeats.

### Genetic diversity

For the microsatellite dataset, we obtained the 12 loci from 14 individuals from the OPHK population and nine individuals from the NCZ population (Table 1). We found no evidence of genotypic disequilibrium after Bonferroni correction. The number of alleles per locus ranged from two to five, and the polymorphic information content per locus ranged between 0.16 and 0.70. For all microsatellite loci, $N_A$ for each population was approximately 3.25, and $N_A$ of the NCZ population was slightly higher than that of the OPHK population. We found low to moderate genetic diversity with a mean $H_O$ of 0.36–0.45 and a mean $H_E$ of 0.34–0.44. Moreover, we found no sign of inbreeding at the loci, with $F_{IS}$ ranging from $-0.283$ ($p = 0.923$) to $-0.214$ ($p = 0.784$) (Table 2).

### Genetic population structure

The simulations performed in POWSIM using our particular microsatellite data and selected sample size showed that the statistical power was sufficient (>90%) to detect genetic substructure if the true $F_{ST} \geq 0.075$ (Fig. S1).

Using multiple approaches, we found no evidence of genetic differentiation between OPHK and NCZ populations. First, significant but low genetic differentiation in the

**Table 3 Characteristics of 12 microsatellite loci in a sample set of two captive Blue-crowned Laughingthrush (*Garrulax courtoisi*) populations.**

| Locus | Repeat motif | Primer sequence | Annealing temperature (°C) | Size range (bp) | $N_A$ | $A_R$ | $H_O$ | $H_E$ | PIC | $F_{ST}$ |
|---|---|---|---|---|---|---|---|---|---|---|
| BCLT_L1 | (TG)6 | F: 5′-CCAAATTCCCCCAGTCCTCC-3′ | 58–60 | 101 | 3 | 3 | 0.261 | 0.241 | 0.222 | −0.008 |
| | | R: 5′-ATGTCAGACACAGCCCGAAC-3′ | | | | | | | | |
| BCLT_L2 | (AC)7 | F: 5′-CCTGCGCATTACCTTGCATC-3′ | 58–60 | 101 | 3 | 4 | 0.087 | 0.086 | 0.082 | −0.019 |
| | | R: 5′-GCAGACACACAGCATTGCAA-3′ | | | | | | | | |
| BCLT_L3 | (GT)7 | F: 5′-ACAAGTCCCACGTGCTTTCA-3′ | 58–60 | 102 | 3 | 4 | 0.261 | 0.300 | 0.262 | 0.137 |
| | | R: 5′-AAACAGTATCCCCTCCCTGC-3′ | | | | | | | | |
| BCLT_L4 | (CT)6 | F: 5′-TGACAAACTCTCCCAAGGCC-3′ | 58–60 | 121 | 3 | 4 | 0.391 | 0.341 | 0.308 | 0.075 |
| | | R: 5′-GCTTTAGCAGGGATGTGGGT-3′ | | | | | | | | |
| BCLT_L5 | (AC)7 | F: 5′-TCCTCAGCTTTCAACCAGGT-3′ | 58–60 | 131 | 4 | 16 | 0.696 | 0.764 | 0.700 | 0.154* |
| | | R: 5′-TCCAGGTGTTGTTCAGTGCA-3′ | | | | | | | | |
| BCLT_L6 | (TA)8 | F: 5′-AAACCAGCCCTCGACCAAAA-3′ | 58–60 | 188 | 5 | 12 | 0.739 | 0.705 | 0.633 | 0.143* |
| | | R: 5′-TCGAGGCTTAATCTGGGTGC-3′ | | | | | | | | |
| BCLT_L7 | (TA)6 | F: 5′-CCCTTCATTAGCCCTGTGCA-3′ | 58–60 | 216 | 3 | 4 | 0.522 | 0.565 | 0.456 | 0.024 |
| | | R: 5′-TTGTGTGTGTGCATGCCATG-3′ | | | | | | | | |
| BCLT_L8 | (GT)6 | F: 5′-AGCAGACCAGAGAGCAACAC-3′ | 58–60 | 238 | 2 | 2 | 0.304 | 0.264 | 0.225 | 0.005 |
| | | R: 5′-TGGCAAAGAAGTTGGGGGTT-3′ | | | | | | | | |
| BCLT_L9 | (TA)6 | F: 5′-TGGAAGCATACACCACACAGA-3′ | 58–60 | 258 | 5 | 5 | 0.565 | 0.538 | 0.484 | 0.028 |
| | | R: 5′-GCATTTTCTTCTTGGCTCTCAGT-3′ | | | | | | | | |
| BCLT_L10 | (TAT)5 | F: 5′-GACAGACACGTGCTTCTCCA-3′ | 58–60 | 137 | 3 | 6 | 0.478 | 0.530 | 0.405 | −0.024 |
| | | R: 5′-GCAGGTCACCTCCTGAACTC-3′ | | | | | | | | |
| BCLT_L11 | (GCA)6 | F: 5′-GGTTCACAGCCTCTGGTCTC-3′ | 58–60 | 184 | 2 | 6 | 0.261 | 0.232 | 0.201 | −0.032 |
| | | R: 5′-AGTTCTGGTTGGGAGTGCTG-3′ | | | | | | | | |
| BCLT_L12 | (TAG)5 | F: 5′-TCCACTTCAGTCCCAGGTCA-3′ | 58–60 | 254 | 3 | 9 | 0.174 | 0.165 | 0.154 | 0.007 |
| | | R: 5′-ATGGCAGTTGGGTTGGAACT-3′ | | | | | | | | |

**Note:**
The average number of different alleles ($N_A$), average allelic richness ($A_R$), mean observed ($H_O$ ± SD), mean expected ($H_E$ ± SD), polymorphism information content (PIC), and genetic differentiation index ($F_{ST}$, * indicates $p < 0.05$) for the OPHK and NCZ populations were estimated for each locus, and values in bold indicate significant deviations from Hardy–Weinberg equilibrium after Bonferroni correction.

microsatellite dataset was revealed by $F_{ST}$ value (0.065, $p = 0.003$). This is because 10 out of 12 loci had non-significant and low $F_{ST}$ values (range, −0.032 to 0.137), with only two exceptions (BCLT_L5: $F_{ST} = 0.154$, $p = 0.007$; BCLT_L6: $F_{ST} = 0.143$, $p = 0.006$) (Table 3). For the STRUCTURE analysis of microsatellite data, we did not find a strong support of a particular number of genetic cluster. The $\Delta K$ estimator (Fig. 1C) suggested that there are most likely two genetic clusters. This is however at odds with the mean posterior probabilities, which has its peak at $K = 1$ (Fig. 1B). Since the $\Delta K$ estimator cannot detect the situation of $K = 1$ and is also meaningless to plot individual assignment of population panmixia, we further did STRUCTURE plot when $K = 2$ (Fig. 1A). It clearly suggests no large genetic difference between the OPHK population and NCZ population. For the PCoA plot based on the 12 microsatellite sites of these two populations, we found no differentiation between the OPHK and NCZ populations in the first principal coordinate but little differentiation in the second principal coordinate (Fig. 2A). Assignment test

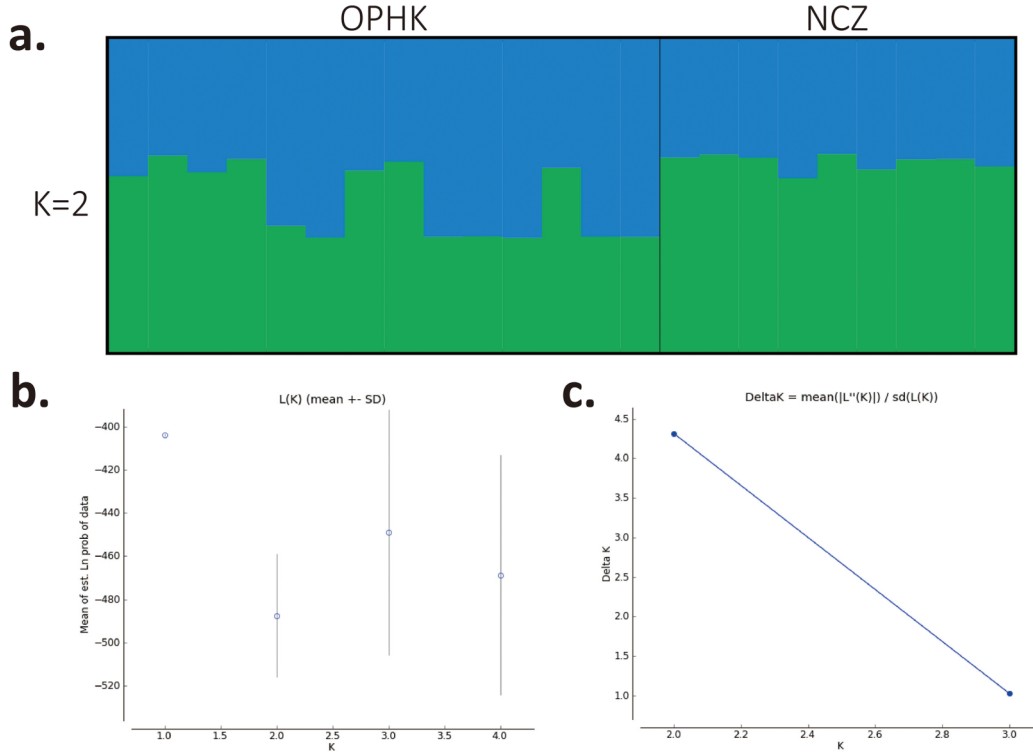

**Figure 1 Population structure results for Blue-crowned Laughingthrush.** (A) Each line represents one individual, and the proportion of population assignment of each individual in relative to each of given genetic cluster is represented by the length of each line. Abbreviations indicate different Blue-crowned Laughingthrush populations (OPHK, Ocean Park Hong Kong; NCZ, Nanchang Zoo). (B) Posterior probability means; Ln P(D) (±SD) increased per K. (C) The second order rate of change of the Ln P(D) with respect to the number of clusters was shown.

results showed that, for the OPHK population, 100% of the individuals (*n* = 14) were assigned to the OPHK population. For the NCZ population, 77.8% of the individuals (*n* = 7) were assigned to the NCZ population, whereas 22.2% of the individuals (*n* = 2, individuals 6,041 and 6,043) were assigned to the OPHK population (Fig. 2B).

For the mitochondrial cytb dataset, we obtained 1,027 bp from each individual (GenBank accession numbers MH423582–MH423604). The average level of genetic diversity was similar between OPHK and NCZ populations (Table 2). Haplotype networks showed that there were five haplotypes among these individuals, and the most frequent haplotype was shared by 18 individuals. Three and one private haplotypes were owned by OPHK and NCZ populations, respectively (Fig. 3).

## Relatedness analysis

The average pairwise relatedness coefficient based on the 12 microsatellite loci within populations ranged from −0.013 to −0.05, and none of these values significantly differed from zero (Table 2). Such a low level of relatedness probably results from the fact that a small proportion of individuals from both populations are closely related (Fig. 4). A dominant proportion of dyads came from unrelated individuals (72.53% in the OPHK population, 86.11% in the NCZ population).

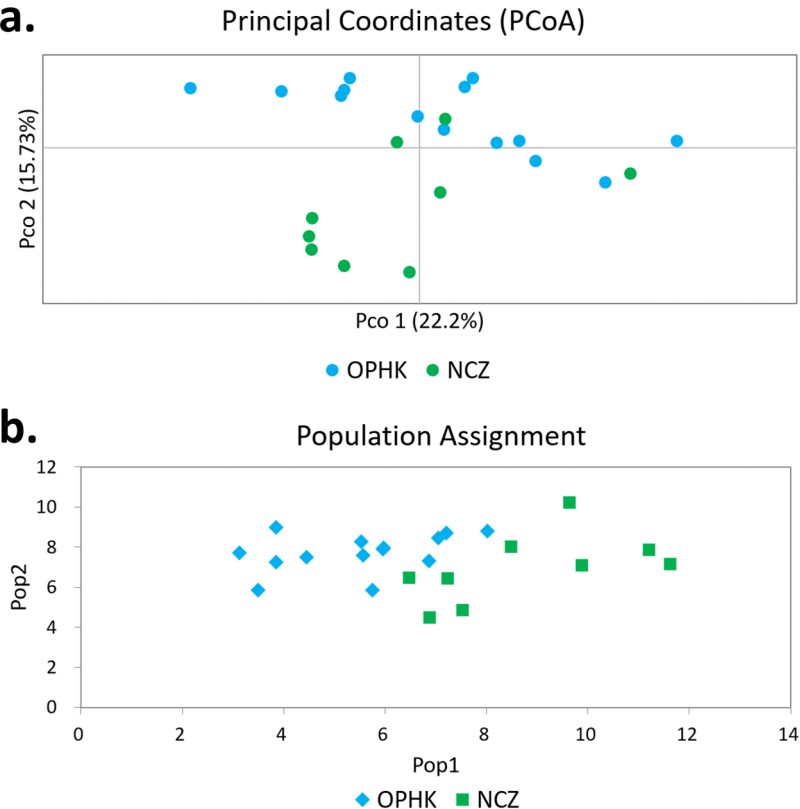

**Figure 2 Principal coordinates analysis and assignment tests.** (A) Principal coordinates analysis results of Blue-crowned Laughingthrush individuals genotyped at 12 microsatellite loci. Different colours represent postulated populations. (B) A biplot of the respective log-likelihood values for individuals from two populations. With log-likelihoods converted to positive values, the lowest value indicates the most likely population of origin. The abbreviations represent the different Blue-crowned Laughingthrush populations (OPHK, Ocean Park Hong Kong; NCZ, Nanchang Zoo).

## DISCUSSION

We provided a set of polymorphic genetic markers for the Blue-crowned Laughingthrush from scanning its genome, which is intended to be a useful genetic tool for efficient conservation of this species. Consequently, we were able to estimate the genetic diversity, population structure, and genetic relatedness of wild and captive populations of this endangered bird species. The obvious next step is to expand the usage of this marker set for other populations in different European and North American zoos (*Wilkinson & Gardner, 2011*; *Wilkinson et al., 2004*). The Global Species Management Plan for the Blue-crowned Laughingthrush has already been approved and facilitates management of zoo populations (*Gardner, 2013*; *World Association of Zoos and Aquariums (WAZA), 2017*). However, conservation genetic information of these populations are not clear. Absence of such information can hinder management and understanding of other genetic problems, such as inbreeding depression (*Frankham, Ballou & Briscoe, 2010*). Besides, it is also necessary to carry out long-term genetic monitoring for the Blue-crowned Laughingthrush in Wuyuan which are facing different conservation challenges (*Zhang et al., 2017*).

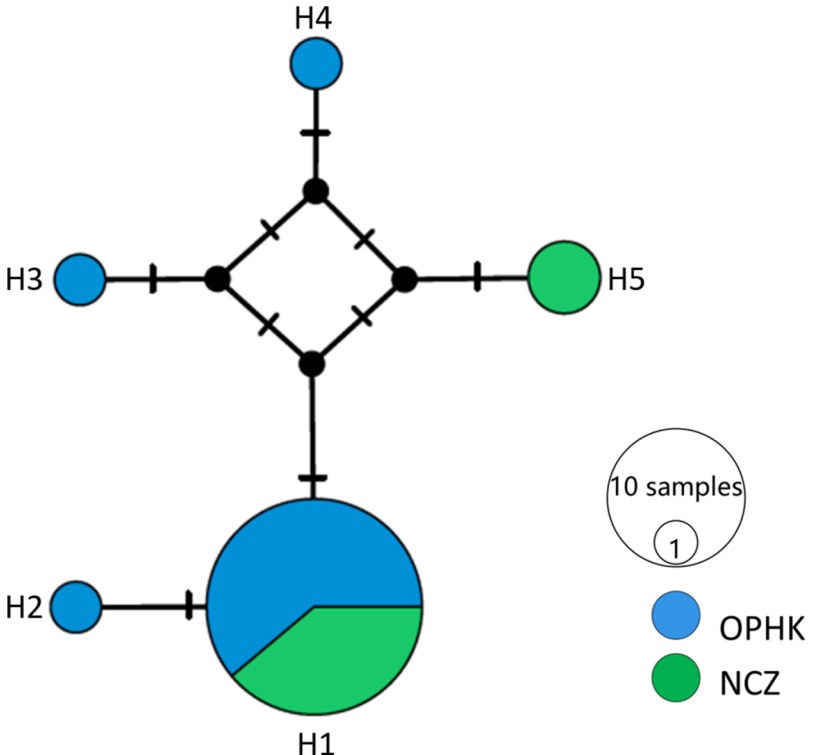

**Figure 3 Haplotype network analysis results for the cytb gene dataset (1,017 bp) of Blue-crowned Laughingthrush.** Black lines on branches indicate the inferred number of mutation steps between haplotypes or ancestral haplotypes. Circle size is proportional to the number of individuals with a particular haplotype. Abbreviations indicate different Blue-crowned Laughingthrush populations (OPHK, Ocean Park Hong Kong; NCZ, Nanchang Zoo).

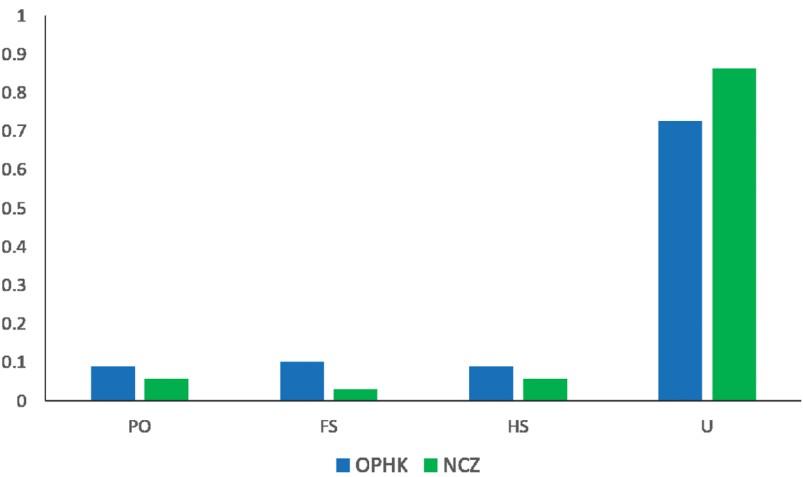

**Figure 4 Pairwise genetic relatedness among Blue-crowned Laughingthrush individuals.** Different colours represent the two study populations (OPHK, Ocean Park Hong Kong; NCZ, Nanchang Zoo). Abbreviations indicate different levels of genetic relatedness (U, unrelated; HS, half-siblings; FS, full-siblings; PO, parent–offspring).

Furthermore, these markers could be broadly applied to conservation genetic studies of *Garrulax* laughingthrushes, a group with great conservation concerns on illegal trade and captive breeding (*Collar & Van Balen, 2013*; *Li, 2009*; *Wu et al., 2012*). It is noteworthy that the utility of crossing species amplification of microsatellites is admittedly feasible (*Dawson et al., 2010*; *Gu et al., 2012*).

The present study reveals first information of genetic diversity, population structure and relatedness of Blue-crowned Laughingthrush. We documented low genetic diversity in two captive populations. Our results showed that genetic diversity of the OPHK population was slightly lower than that of the NCZ population, but there was no significant difference between them. Unexpectedly, we found no evidence of inbreeding among closely related individuals in both captive populations. For the OPHK population, the best explanation is that this captive population likely originated from different wild sources (as described in the section below), and this might provide genetic rescue that prevented loss of genetic diversity. The results of the NCZ population may be related to the short amount of time (<10 years) that this population has been captive. And our result of relatedness estimator may also support these results, most value of the pairwise relatedness ($R_{QG}$) are negative in both OPHK population and NCZ population (Table S2). The more negative value means the more confident of unrelated of that two individuals, and it further indicate the detection of recent immigrants that carry novel alleles (*Konovalov & Heg, 2008*; *Queller & Goodnight, 1989*). Another factor that must be considered is the statistical power of the microsatellite data. Our simulation showed the power of this marker set was high (>90%) if the true $F_{ST} \geq 0.075$. However, the actual average pairwise $F_{ST}$ was 0.065. Such weak population differentiation may cause the underestimation of the inbreeding and relatedness (*Liu, Keller & Heckel, 2011*, *2013*).

We found a weak signal of genetic differentiation between OPHK and NCZ populations based on different methods. Although it may be some arguments about the chosen of the number of clusters in STRUCTURE, since the $\Delta K$ helps in identifying the correct number of clusters in most situations, it cannot find the best $K$ with the situation a panmictic population (*Evanno, Regnaut & Goudet, 2005*). The lack of strong population subdivision may be the most consensus result because of the highest mean posterior probabilities at $K = 1$ (Fig. 1B). Indeed, this result is consistent with other methods, such as PCA, that is, largely overlapped in PCo1 (Fig. 2A), and assignment tests. We inferred that these two populations only have slightly different genetic components. For instance, assignment analysis revealed that two individuals of the NCZ population were inferred to be from the OPHK population; this result, which may be due to the low sample size of the NCZ population, further implies that higher genetic variation is present among individuals in the OPHK population. Moreover, mitochondrial DNA data analysis also supported this result that most individuals shared the same haplotype. However, both the OPHK and NCZ populations had private haplotypes. One parsimony explanation is that at least some individuals of the OPHK shared the most recently ancestor with the NCZ population, in other words, from Wuyuan wild origin.

However, the population genetic analysis performed in this study, that is, HWE tests and *F*-statistics, (e.g. inbreeding coefficient and fixation index $F_{ST}$) were measured based

on a very small sample size for each captive population (14 and seven individuals, respectively) and genotyped at 12 loci. Because both HWE tests and $F$-statistics lie the assumption of an infinite population size (*Guo & Thompson, 1992*; *Weir & Cockerham, 1984*), our dataset using rather small population size (<20) may lead to ascertainment bias and an overestimation of $F$-statistics (*Kalinowski, 2005*; *Willing, Dreyer & Van Oosterhout, 2012*). Thus the biological interpretation of these results should be used with great caution. The relative precise estimates can be obtained by incorporating more individuals (*Kalinowski, 2005*) and/or more genetic markers (*Willing, Dreyer & Van Oosterhout, 2012*). If increasing sample size is not possible, usually the case for extremely rare and endangered species, it is more feasible to genotype a small number of individuals at a larger number of genome-wide markers developed with high-throughput sequencing (*Davey et al., 2011*; *Luikart et al., 2003*).

Together, our results do not support previous hypothesis that captive OPHK individuals may belong to the subspecies *simaoensis* (*Wilkinson & Gardner, 2011*; *Wilkinson et al., 2004*). However, we are unable to exclude a single source of individuals in the OPHK population. Consequently, the safe conclusion is that individuals from Wuyuan, perhaps also Simao populations have contributed to the OPHK population. Our results further calls the re-evaluation of the subspecies status of *simaoensis*. Some taxonomic treatment argued that *simaoensis* may not be a valid subspecies, as its diagnostic character, that is, yellowish-grey breast-band sometimes also presents in Wuyuan wild population (*Collar, Robson & Sharpe, 2019*). Because the population status of *G. c. simaoensis* is unclear and it may have already been regionally extirpated (*Wilkinson & He, 2010a*), genomic approaches applied on genotyping a handful number of museum specimens of putative *G. c. simaoensis* has great potential to resolve this issue.

## CONCLUSIONS

With the new set of markers we proposed, we estimated genetic diversity, structure and relatedness of captive Blue-crowned Laughingthrush populations for the first time. Information and permanent genetic resources obtained from this study can benefit effective ex situ and in situ conservation efforts to recover bird species from the brink of extinction.

## ACKNOWLEDGEMENTS

We are grateful to Dr. Emilio Pagani-Núñez for his comments on a previous version of this manuscript. We thank Mallory Eckstut, PhD, from Liwen Bianji, Edanz Editing China for editing the English text of a draft of this manuscript. Two anonymous reviewers and Prof. Michael Wink provided useful comments on revising the paper.

### Funding

This work was supported by a grant from the Ocean Park Conservation Foundation, Hong Kong, China (No. BD03_1617) to Yang Liu. The funders had no role in study design, data collection and analysis, decision to publish, or preparation of the manuscript.

## Grant Disclosure

The following grant information was disclosed by the authors:
Ocean Park Conservation Foundation, Hong Kong, China: BD03_1617.

## Competing Interests

Guoling Chen is a master student and Yang Liu is an Associate Professor in Sun Yat-sen University. Chenqing Zheng and Xu Yang are bioinformaticians employed by Shenzhen Realomics Biological Technology Ltd. Nelson Wan is employed by Ocean Park Corporation Hong Kong. Daoqiang Liu is employed by Nanchang Zoo. Vivian Wing Kan Fu and Yat-tung Yu are conservationists in The Hong Kong Bird Watching Society, a bird conservation NGO.

The authors declare that they have no competing interests.

## Author Contributions

- Guoling Chen performed the experiments, analysed the data, contributed reagents/materials/analysis tools, prepared figures and/or tables, authored or reviewed drafts of the paper.
- Chenqing Zheng analysed the data, contributed reagents/materials/analysis tools.
- Nelson Wan conceived and designed the experiments, performed the experiments, sampling.
- Daoqiang Liu performed the experiments, sampling.
- Vivian Wing Kan Fu conceived and designed the experiments.
- Xu Yang contributed reagents/materials/analysis tools.
- Yat-tung Yu conceived and designed the experiments.
- Yang Liu analysed the data, contributed reagents/materials/analysis tools, prepared figures and/or tables, authored or reviewed drafts of the paper, approved the final draft.

## Animal Ethics

The following information was supplied relating to ethical approvals (i.e. approving body and any reference numbers):

All the animal operations were approved by the Institutional Ethical Committee of Animal Experimentation of Sun Yat-sen University (2005DKA21403-JK) and strictly complied with the ethical conditions by the Chinese Animal Welfare Act (20090606).

## Data Availability

The raw data of microsatellite genotype are provided in Table S1.

The raw data of the cytb gene is included in Supplemental Files and taken from GenBank accession numbers MH423582–MH423604.

## Supplemental Information

Supplemental information for this article can be found online at http://dx.doi.org/10.7717/peerj.6643#supplemental-information.

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
