# Peer review of "Low genetic diversity in captive populations of the critically endangered Blue-crowned Laughingthrush (Garrulax courtoisi) revealed by a panel of novel microsatellites"

_PeerJ, doi:10.7717/peerj.6643_

## Round 0.1 · original submission · Major Revisions

Dear authors
please read the reviews carefully. For your revision, you need to take the recommendations of reviewer 2 seriously. He will see your revision.
Kind regards
Michael Wink
AE

Reviewer 1 ·

Basic reporting

Apart from some minor language issues (see below), language is appropriate.
Cited literature is appropriate.
The first figure is superfluous, one table needs additional information.
There are no clear hypotheses formulated, but the authors present what they announce.

Experimental design

Appropriate methods were used and are sufficiently described in the MS apart from minor issues (see below).

Validity of the findings

Some conclusions tend to reach a bit too far (see below), otherwise presentation and interpretation of results are good.

Additional comments

Lines 120 and 122 give different years.
Line 145 makes no sense: One cannot (directly) sequence a draft genome. What did you sequence? Was the outcome an assembly worth being called a draft genome? Which sample was used for the genome or were all samples pooled?
Line 175: why arbitrarily?
Lines 181 ...: consistently use l OR L for liter(s)
Lines 214, 226, 255: consistently use subscripts
Lines 342-4: Why are you so confident that your markers work in other or even all Garrulax (s.l. or s.s.?) species?
Line 430: Are museum specimens available? Why were they not used? Are they too old to provide DNA for MS studies?
Do you really need a photo = Figure 1?
Figure 4 is not a haplotype network.
Table 1: What is the orientation of the primer sequences?

Reviewer 2 ·

Basic reporting

The english is acceptable for publishing.

The reference list is sufficient but you will see that I recommend shortening the manuscript substantially which may result in removing references (see general comments).

The article structure is professional. Again, I recommend shortening the manuscript o the basic findings and this may allow fusing results and discussion (iff this is covered by the editorial rules).

As outlines in my general comments, I feel that the manuscript has the structure of a primer Note which is a technical paper. Thus, hypothesis stating and testing is limited and the paper is rather descriptive than hypothesis driven.

Experimental design

The motivation of the paper is justified.

The rather descriptive and technical character of the work allows just limited hypothesis building.

There are analytical flaws:

Number of samples:
Studies on conservation genetics and endangered species usually deal with low samples sizes and one cannot overcome this problem (this paper: OPHK = 14, NCZ = 9)! However, one has to be aware that a low sample size affects the accuracy of parameter estimates and the statistics behind it (as a rule of thump: 20 spec. minimum to get reliable values). This counts for the diversity and divergence analyses presented in tables 1 and 2. Thus, the authors should indicate this problem throughout the manuscript and scale down their interpretations, e.g. related to deviations from Hardy-Weinberg equilibrium or indirect conclusions about inbreeding (tables 1, 2 and text related to theses tables).

Bayes' analyses (STRUCTURE):
Please revise the presentation of these results both in the text (e.g. results: lines 306-311) and figure 2. Figure 2 shows in panel (a) the STRUCTURE bar plot (Please indicate clearly in the caption that the three plots refer to K2, K3, K4 and the two populations under consideration). These bar plots (particularly the upper one) indicate admixture (F1 like) between two genetic clusters. However, I miss any interpretation of this. I also think that the additional bar plots putatively representing results for K3 and K4 are meaningless and should be removed simply to keep the manuscript as concise as possible. Presentation of bar plots for K other than this considered to be the most likely one (e.g. as determined by the Evanno method) is only meaningful if they indicate additional and biologically interpretable substructures. Please also double check panels (b) and (c). Panel (b) is labeled with DeltaK... but the respective caption says Posterior probability (Ln P(D)). Panel (c) is labeled with L(K) but the respective caption says Delta K. I ask the authors to explicitly present a graph showing the mean Ln P(D) and to consider and discuss the possibility of K=1 which is not detectable by the Evanno-method but is a possible analysis outcome.

Validity of the findings

The manuscript has the character of a Primer Note rather than a research paper although microsatellites result from whole draft genome sequencing. This feeling is also due to the lack of data from free living populations which makes the results difficult to interpret.

Additional comments

Chen et al. develop a new set of microsatellites for the critically endangered Blue-crowned Laughingthrush. The motivation of the study is to provide a conservation genetics tool for monitoring genetic diversity and divergence that was exemplarily applied to two captive populations. The authors discuss their results in the context of current conservation strategies for the bird. The study is more than justified and gets is significance from the fact that genetics can substantially contribute to monitoring activities in combination with other disciplines, e. g. field work or ecology. This makes the content of the manuscript publishable and interesting for the scientific community particularly for people working in the field of ornithology and/or conservation.

However, I cannot support the publication of this manuscript in its current form. This primarily arises from what I said in respect to Experimental design and Validity of the findings. Furthermore, the paper has a strong focus and does not provide information for a broad scientific community in biosciences. Apart from this strong focus, the conclusions drawn about the population origin and the genetic status (e.g., inbreeding) of the two captive populations are limited (see discussion and conclusions part). I did not lose the impression that the content of the paper should be presented in the style of a Primer Note with an extended discussion covering the specific aspects of conservation. Consequently, I recommend substantial shortening the text just focussing on the main aspects, i.e. the need for this molecular tool, the basic properties of the loci, the main results and interpretations of the study of these two captive populations. I also think that the manuscript might get more attention in a journal with particular focus on ornithology.

I recommend addressing the aspects raised particularly in the paragraphs on "Experimental Design" and "Validity of the findings" and give some additional comments below. Once all these aspects are addressed I may be able to guaranty correctness of the study and may support publication.

Additional comments:

Figure 4: Please double check Figure 4. The document that I downloaded for review shows a bar chart rather than a network and the caption appears to not fit with what is shown there.

Comparison between diversity and divergence between different species and marker sets (Table 3):
Comparing between independent estimates of mitochondrial and microsatellite diversity between different species and different marker sets is dificult and under many circumstances even impossible and misleading. This is because different species have different life cycles and histories and one has to argue why a comparison is meaningful. Further, the comparison refers to non-orthologous loci which further limits the validity of the comparison. Genetic drift and/or selection acts differentially on the freely recombining nuclear loci and, therefore, affects locus specific estimates.

Discussion: This part of the manuscript provides a lot of general statements not least about the importance of conservation genetics (there is no doubt about). This makes the discussion overly long and partly unfocussed. The authors should rather concentrate on how their marker set could be incorporated in monitoring activities of free living populations and/or how their results on captive populations can be used for the management. Is it feasible to use the captive animals for reintroduction in the wild? If yes, I suggest an extensive discussion about putative problems arising from releases of captive animals in the wild, e.g. outbreeding depression. I argue that the scientific value of this manuscript greatly depends on the potential use of the data in a management plan (that could be developed or outlined here).

---

## Round 0.2 · Major Revisions

Dear authors

You can see that Reviewer 2, who is very competent, is not happy with your revision. Try to implement the suggestions, otherwise we cannot accept your ms.

Kind regards

Michael Wink
Academic editor

Reviewer 1 ·

Basic reporting

-

Experimental design

-

Validity of the findings

-

Additional comments

The authors intensively revised the MS, following our recommendations.

Only language needs some smoothing, e.g.:
Line 373: two populations
Figure 1: "Each line represents one individual, and the possibility of individual that assigned to the given genetic cluster is represent by the length of each line." Represented? Is possibility the right English term?

Reviewer 2 ·

Basic reporting

The first revision of this article is again written in an acceptable English for publication with an appropriate reference list. I appreciate that the authors followed my suggestion to substantially shorten the manuscript; particularly the discussion. I feel that the length of the manuscript is more appropriate with respect to results than the original submission.

Experimental design

I am disappointed about the way how the authors addressed my previous comments on several aspects of the genetic analyses.

First, I asked the authors to explicitly comment on the low sample size in respect to Hardy-Weinberg-Equilibrium (HWE) tests and interpretations about signs of inbreeding. Instead of doing so, they refer to the POWSIM test in respect to divergence (FST). Note, this was part of the original manuscript and my comment did NOT refer to this well chosen analytical tool. Furthermore, the authors discuss the impact of low sample size in assignment tests. I renew my comment and kindly ask the authors to interpret their results on HWE and inbreeding, e.g., in results under the headline genetic diversity.

Second, I explained in detail my concerns on the interpretation of the STRUCTURE results. Apart from correcting and improving the label in Figure 1 (Figure 2 of the original submission), the authors did not biologically interpret the clear signs of admixture but added methodological information. This was not meant by my comments. I renew my concern and state that STRUCTURE results could be consistent with K=1 given the high mean lnPD value (with minor SD) as shown in Figure 1 B of the revised version. I again ask to provide a biological interpretation for admixture which puts the statement about divergence between OPHK and NCZ in a slightly different context. I suggest that divergence is extremely low (FST 0.065; line 294 of the revised manuscript) and this low level of divergence is not appropriately reflected by STRUCTURE analyses but only (slightly) by PCA. I repeat again that the Evanno method cannot detect K=1 but K=1 is a possible outcome of Bayesian clustering that has to be considered. In addition, I asked to provide an explanation why bar plots for K=3 and K=4 are presented. I still argue that they do not add to the topic.

Validity of the findings

The validity of the findings are presented more focussed in the new (shortened) manuscript and become clearer.

Additional comments

If my concerns listed under experimental design are seriously and explicitly addressed based on scientific arguments, I will support publication of this manuscript. I currently feel that the authors did not provide convincing arguments to reject my suggestions from the first round of review.

---

## Round 0.3 · accepted · Accept

Dear authors

Congratulations! Your revision is appropriate and thus we can accept your manuscript.

Greetings,
Michael Wink
Academic Editor

# Reviewer 2 ·

Basic reporting

This manuscript is in the second round of revision which required only few additional changes in respect to some scientific issues. The incorporated changes come in a clear and professional English and do not destruct the professional article structure. The new sentences explain results and hypotheses in in competent way.

Experimental design

The changes required in this seconds round of review did not refer to the experimental design or any methods used in the original paper/first review.

Validity of the findings

The new interpretations now correctly reflect the validity of the findings.

Additional comments

Thanks for the competent revision. I consider this version of the manuscript as a nice contribution to conservation.